# Deep Metric Learning Using Negative Sampling Probability Annealing

**DOI:** 10.3390/s22197579

**Published:** 2022-10-06

**Authors:** Gábor Kertész

**Affiliations:** John von Neumann Faculty of Informatics, Obuda University, 1034 Budapest, Hungary; kertesz.gabor@nik.uni-obuda.hu

**Keywords:** negative sampling probability annealing, triplet mining, deep metric learning, triplet network

## Abstract

Multiple studies have concluded that the selection of input samples is key for deep metric learning. For triplet networks, the selection of the anchor, positive, and negative pairs is referred to as triplet mining. The selection of the negatives is considered the be the most complicated task, due to a large number of possibilities. The goal is to select a negative that results in a positive triplet loss; however, there are multiple approaches for this—semi-hard negative mining or hardest mining are well-known in addition to random selection. Since its introduction, semi-hard mining was proven to outperform other negative mining techniques; however, in recent years, the selection of the so-called hardest negative has shown promising results in different experiments. This paper introduces a novel negative sampling solution based on dynamic policy switching, referred to as negative sampling probability annealing, which aims to exploit the positives of all approaches. Results are validated on an experimental synthetic dataset using cluster-analysis methods; finally, the discriminative abilities of trained models are measured on real-life data.

## 1. Introduction

Image-based instance re-identification, one- and few-shot learning, and image similarity analysis are popular fields of computer vision research, with applications in vehicle recognition [1,2] and tracking [3], and facial recognition and identification [4]. Solutions based on deep learning [5] use a dimensionality-reduction technique to transform observations into an embedded space where distance represents similarity—this is referred to as metric learning.

Over recent decades, multiple machine learning approaches have been studied, starting with the work of Bromley et al. who introduced the Siamese architecture [6] for signature verification. Recently, in the era of deep learning, applications in facial recognition drove researchers to reach peak performance; DeepFace [7], FaceNet [8], and OpenFace [9] represent deep-metric-learning-based solutions for facial identification, applying novel solutions, such as triplet loss and triplet mining.

The application of triplet loss—as an error function—as a replacement for the contrastive loss of the Siamese net redefines the structure of the problem as well. The Siamese net consists of two inputs, and loss is calculated based on the distance of the embedded vectors; for same-class elements, loss is proportional to distance, and the opposite is true for elements of different classes.

The training process differs from supervised learning where, in an epoch, every sample is used for training. When training a Siamese-architectured network, applying every single sample would produce a total of $\frac{n\cdot (n-1)}{2}$ for *n* (number) total training samples. In the case of deep learning, where a large number of training samples are used, the exponential growth of training steps is difficult to handle. Another problem in this scenario is the imbalance of positives and negatives: by increasing the number of classes, the number of pairs from different classes increases when compared with the number of same-class pairs. The solution—of course—is to select both positive and negative pairs in the same number during training.

For triplet nets, three samples are used during training: a so-called anchor element used as the basis, along with a positive and a negative pair, formally given as T={a,p,n}. These elements—the triplet—are then used to calculate the loss using the triplet loss [8]:(1)$$L=\max(0,m+d^{+}-d^{-}),\;$$
where $d^{+}$ and $d^{-}$ refer to the positive and negative distance, respectively, defined as
(2)$$\begin{array}{c} d^{+}=d\left(a,p\right),\\ d^{-}=d\left(a,n\right), \end{array}$$

d(·,·) representing the distance function, and *m* is the margin.

Similarly as above, a training that includes all possible triplets would be inefficient and computationally heavy, so triplets are selected such that the training is effective.

Triplet mining is the method of selecting a batch of triplets where the training would result in a non-zero loss; of course, other policies are often taken into account. In theory, a high loss could be measured for a given *a* having the furthest *p* and the closest *n* to form as a triplet; however, experiments on real-life data show that the result is inflexible clusters and the classification performance of the models on unseen samples is low [10,11].

Therefore, other strategies for positive and negative sampling are often used. For selecting *p*, so-called *easy* sampling is popular, where *easy* refers to the low computational cost of random selection. On the other hand, *hard* positive sampling refers to filtering and selecting a positive which is further than the other candidates.

For negative sampling, after *a* and *p* are known, different policies are available. The previously mentioned *easy* mining is a random selection; however, it is not a good choice: because of (Equation 1), if the selected *n* results in a $d^{-}$ where $d^{-}>d^{+}+m$, then the resulting L is zero. Therefore, negative sampling should always consider those *n* candidates, where L is non-zero [12]. There are also choices on how to select negative pairs for the triplet. Random hard mining refers to randomly selecting *n* from candidates where the resulting loss is non-zero. Hardest mining is selecting *n* with the minimal $d^{-}$, i.e., the closest one. Another interesting sampling technique is semi-hard mining, where *n* is randomly selected from those elements that result in non-zero loss, but are in the marginal distance.

Unfortunately, none of the above sampling techniques provide general, solid methods for deep metric learning; multiple approaches were investigated to improve performance.

### Related Work

The paper on FaceNet introduced triplet loss [8], a similar solution presented in a paper by Hoffer et al. [13]; since these, multiple extensions, alterations, and optimization techniques, along with some interesting use cases were revealed.

Amos et al. showed [9] that triplet mining is useful, and can be performed efficiently. Online triplet mining—the method where triplets are selected right before the training step—is performing remarkably better than its counterpart (*offline* mining), where embeddings are unaffected by the last steps [14].

Wu et al. showed [12] that sampling matters, and it is even more significant than the selected loss function—or the hyperparameters of the selected function—in achieving high performance. The first sampling method was proposed in the original paper by Schroff et al. [8]: semi-hard negative sampling showed great potential in converging to optimal clustering. A similar approach (although for Siamese nets) was proposed by Simo-Serra et al. [15], where positive pairs and negative pairs were sorted by loss in descending order, and used for training accordingly. Harwood et al. [16] proposed an online mining method based on semi-hard sampling of negatives. Hermans et al. [17] showed that the use of hard positive sampling is effective in clustering same-class samples, although it is also demonstrated that the method is unstable and not necessarily applicable to all kinds of data. Recently, Xuan et al. [18] showed that hard negative sampling at early stages leads to stucking in local minima, resulting in a suboptimal model. Kalantidis et al. [19] proposed a method for synthetic hard negative mixing to gain advantages of hard negative mining, without the drawbacks.

Different loss function alternatives were also proposed and analyzed: first and foremost, magnet loss [20] is a computationally hard approach with high memory costs continuously analyzing the clusters. Due to the frequent stops during the training process, the magnet loss method might show high performance, but it is quite inefficient. Wang et al. [21] introduced the ranked list loss to deal with sampling issues, and used data from *non-trivial* data points as well. Alternatives of the triplet loss, such as exponential triplet loss [22] and lossless triplet loss [23], have also been shown to be promising alternatives. It is also worth mentioning that a recent study showed that classical methods show similar performances to state-of-the-art loss functions [24].

A different viewpoint of the problem concentrates on the fact that metric learning is performed on Euclidean space; however, loss is based on non-Euclidean distances [25]. Novel solutions are based on a special type of Riemannian manifold, e.g., the Grassmann manifold or the Stiefel manifold [26]. Multiple studies have concluded that the application of metric learning in nonlinear structures shows promising results in face recognition [27,28,29].

In some cases—when applicable—pre-training or transfer learning applied to the backbone model is proven to be useful for deep metric learning applications [30,31,32]. Modern facial recognition solutions—e.g., SphereFace [33,34]—propose solutions based on angular marginal loss, which inspired multiple other works on the same topic, e.g., CosFace [35]. Another interesting application in visual sensors is for object tracking: recently, metric-learning-based trackers showed great performance [36,37]. The state-of-the-art results on applying the SiamRPN++ [38] tracker further fuel the research in deep metric learning.

In this paper, a method of cluster analysis is introduced; based on the observations, a novel negative sampling method is proposed and evaluated. The main contributions of the current study are:Formal description and summary of negative sampling methods;Cluster-analysis methods formally introduced and applied during different experiments on a synthetic dataset;Introduction of the negative sampling probability annealing algorithm for negative selection, with evaluation of results on synthetic dataset as well as actual data; it can be concluded that—while the best-performing models behave similarly—in general, NSPA and random hard sampling seem to outperform other approaches.

The structure of the paper is as follows: in Section 2.1, the triplet network is introduced in formal terminology, including some of the sampling methods. In Section 2.2, the cluster analysis and the properties are presented. Section 2.3 describes the novel method on negative sampling. Section 3 describes the implementation and presents the results; finally, in Section 4, a discussion of the results is presented.

## 2. Methodology

In this section, inspiration and methods are presented in detail.

### 2.1. Triplet Network

As previously defined, a training triplet can be formalized as
(3)$$\begin{array}{c} T=\{a,p,n\},\\ a,p,n\in X,\; \end{array}$$
where *a* represents the anchor, *p* is the positive pair, and *n* is the negative pair, all coming from the dataset, *X*. Function f(·) represents the class of each sample as
(4)$$\begin{array}{c} f:X\to Y,\\ \forall x\in X,\exists y\in Y:f(x)=y,\; \end{array}$$
where *Y* represents the set of classes. Based on this, the positive pair can be defined as
(5)$$\begin{array}{c} \forall a\;\exists p:f(a)=f(p),a\neq p.\; \end{array}$$

Similarly, *n* can be given as
(6)$$\begin{array}{c} \forall a\;\exists n:f(a)\neq f(n).\; \end{array}$$

This case is—of course—an example of easy negative sampling: any randomly selected negative sample fits the criteria. Given $n_{H}$ as a random hard negative, where the loss is non-zero:(7)$$\begin{array}{c} \forall a,p\;\exists n_{H}:f\left(a\right)\neq f\left(n\right),\;d\left(a,n_{H}\right)<d\left(a,p\right)+m\\ a,p,n_{H}\in X,\;m>0.\; \end{array}$$

Semi-hard sampling can be similarly defined as
(8)$$\begin{array}{c} \forall a,p\;\exists n_{SH}:f\left(a\right)\neq f\left(n\right),\;d\left(a,p\right)<d\left(a,n_{SH}\right)<d\left(a,p\right)+m\\ a,p,n_{SH}\in X,m>0.\; \end{array}$$

Finally, the hardest negative is the one which maximizes the loss for a given *a* and *p*:(9)$$\begin{array}{c} n_{H*}=\left\{\arg\min{\left\{d\left(a,x\right)\right\}}_{x}^{X}|f\left(a\right)\neq f\left(x\right)\right\}.\; \end{array}$$

In the following, training is performed by selecting random anchors and applying easy positive mining—i.e., randomly selecting a positive pair, given as
(10)$$\begin{array}{c} \forall a:a\in_{R}X;\\ \forall p:p\in_{R}X,f\left(a\right)=f\left(p\right).\; \end{array}$$

On the negative sampling method, if random hard, semi-hard, or the hardest mining is applied, *n* is one of the following:(11)$$n\in \{n_{H},n_{SH},n_{H*}\}.\;$$

Figure 1 presents triplets and negative sampling possibilities.

### 2.2. Cluster Analysis

To observe the effects of different sampling methods, cluster properties are observed on a synthetic dataset during triplet-loss-based metric learning. The following properties were defined and implemented and measured during training, Section 3 presents the findings.

For cluster analysis [39,40], we define the centroid, $c_{y}$, of class, *y*, as
(12)$$\begin{array}{c} c_{y}=\left[c_{y}\left(1\right),c_{y}\left(2\right),\cdots ,c_{y}\left(n\right)\right],\\ c_{y}\left(i\right)=\sum_{x_{y}\in X_{y}}x_{y}\left(i\right)/\left|X_{y}\right|,\\ X_{y}=\left\{x\in X|f\left(x\right)=y\right\},\; \end{array}$$
where *n* represents the dimensionality of the data points, *y* stands for the selected class, and $X_{y}$ represents the set of elements of class *y*. Centroid, $c_{y}$, is given by selecting the averages of the values in each dimension.

One of the measured properties is the average distance of centroids, which simply represents
(13)$$\begin{array}{c} \forall y_{1},y_{2}\in {\left[Y\right]}^{2},\\ \mathrm{avgCentroidDistance}=\sum_{y_{1},y_{2}\in {\left[Y\right]}^{2}}d\left(c_{y_{1}},c_{y_{2}}\right)/\left|{\left[Y\right]}^{2}\right|,\; \end{array}$$
where ${\left[Y\right]}^{2}$ represents the set of all combinations with two elements of *y* classes without repetition.

For each class, the radius of the cluster is given as the distance of the centroid and the furthest class member, formally described as
(14)$$r_{y}=\max_{x_{y}\in X_{Y}}\left\{d\left(x_{y},c_{y}\right)\right\}.\;$$

Another measured property is the average cluster radius, which is given as
(15)$$\begin{array}{c} \forall y\in Y,\\ \mathrm{avgClusterRadius}=\sum_{y\in Y}r_{y}/\left|Y\right|.\; \end{array}$$

Figure 2 shows a sample for cluster centroid and the defined radius.

Based on this definition of a cluster, we can conclude that, in this case, all elements of the cluster will be inside the cluster, as $\forall x_{y}\in X_{y}:\;d\left(x_{y},c_{y}\right)\leq r_{y}$. An interesting property is the number of other elements, which is given as
(16)$$\begin{array}{c} n_{y}=\left|\forall x\in X,x\notin X_{y}:\;d\left(x,c_{y}\right)\leq r_{y}\right|,\\ q_{y}=\frac{n_{y}}{n_{y}+\left|X_{y}\right|},\\ \mathrm{avgPctOfNegativesInCluster}=\sum_{y\in Y}q_{y}/\left|Y\right|,\; \end{array}$$
where $m_{y}$ gives the number of negatives in cluster *y*, and $q_{y}$ gives the ratio of negatives in comparison to all elements in the cluster. Similarly, the proportion of negatives in marginal distance can be given based on margin *m* by having $d\left(x,c_{y}\right)\leq r_{y}+m$ as the filter.

Further properties of the clusters can be given solely on the distances of data points, e.g.,
(17)$$\begin{array}{c} \forall x_{1},x_{2}\in {\left[X\right]}^{2},\\ \mathrm{avgDistanceBetweenElements}=\sum_{x_{1},x_{2}\in {\left[X\right]}^{2}}d\left(x_{1},x_{2}\right)/\left|{\left[X\right]}^{2}\right|, \end{array}$$
gives the average distance between any two elements. Similarly, the mean distance between positive elements can be given by measuring the distance between all same-class elements in all classes, *y*, and returning with the average.

The maximal distance between any pair of elements of a given class can be formalized as
(18)$$\begin{array}{c} \forall x_{y_{1}},x_{y_{2}}\in {\left[X_{y}\right]}^{2},\\ g\left(y\right)=\max{\left\{d\left(x_{y_{1}},x_{y_{2}}\right)\right\}}_{x_{y_{1}},x_{y_{2}}}^{{\left[X_{y}\right]}^{2}},\\ \mathrm{avgDistanceOfFurthestPositives}=\sum_{y\in Y}g\left(y\right)/\left|Y\right|, \end{array}$$
where function g(·) gives the distance of the furthest pair for a given class, *y*.

An other interesting measurable property is the average distance of the closest negative for each data point. Formally given,
(19)$$\begin{array}{c} h\left(x\right)=\min{\left\{d\left(x,x^{-}\right)\right\}}_{x^{-}}^{X}|f\left(x\right)\neq f\left(x^{-}\right),\\ \mathrm{avgClosestNegatives}=\sum_{x\in X}h\left(x\right)/\left|X\right|, \end{array}$$
where function h(·) represents the distance to the closest different classed element for a given input x∈X.

At the beginning of the training process, the clusters are not separable, and the elements are scattered in the embedded space. Therefore, the average radius, the average positive distance, the average distance of closest negatives, and the average distance of the furthest positives are not informative by themselves. These distances, in contrast to the average distance between all elements, are informative—i.e., we expect the positive distances to decrease compared with the dynamic of average distance. A normalized value of these properties are hereby defined, e.g.,
(20)$$\mathrm{normDistanceOfFurthestPositives}=\frac{\mathrm{avgDistanceOfFurthestPositives}}{\max\{\mathrm{avgDistanceBetweenElements},\epsilon \}},$$
where the denominator is non-zero, thus the threshold ϵ, which is selected as $10^{-5}$.

Further details about implementation can be found in Section 3.

### 2.3. Negative Sampling Probability Annealing

By reviewing the relevant literature and analyzing the results of the cluster analysis on synthetic data (presented in detail in Section 4), a method of negative sampling policy switching is proposed.

The main ideas are as follows:Random hard sampling is efficient, and converges at the start;Semi-hard sampling performs better than the other methods;Sampling the hardest negative has potential, but only in later phases.

Therefore, as a naïve approach on sampling switching, random sampling should be used at the beginning, which should be followed by semi-hard mining, and finally hardest mining should be used. This approach would be unnecessarily rigid however, and the optimization of hyperparameters of phase boundaries seems to be key.

The evaluation of a random sampling policy-switching method showed [41] that both the drawbacks and benefits of each technique are combined in the results, while the method itself looks promising.

As a middle-ground between the abovementioned approaches, a probability-based method is proposed where probabilities of different sampling policies change during the training process.

Formally, let *H* represent random hard negative sampling, SH stand for semi-hard negative sampling, and H∗ stand for hardest negative selection. For a uniform random policy selection, the probabilities of using each method are given as
(21)$$\begin{array}{c} P\left(H\right)=\frac{1}{3},\\ P\left(SH\right)=\frac{1}{3},\\ P\left(H*\right)=\frac{1}{3}, \end{array}$$
or by defining P=[P(H),P(SH),P(H∗)]
(22)$$P=\left[\frac{1}{3},\frac{1}{3},\frac{1}{3}\right].$$

In the proposed annealing method, P=[1,0,0] initially, and occasionally the values are updated by increasing both P(SH) and P(H∗) with $step_{SH}$ and $step_{H*}$, while having
(23)∑P(i)=1,
therefore continuously decreases the value of P(H). Detailed behavior is explained in Algorithm 1.

The fundamental behavior is that the probability update method defined in Algorithm 1 is called during training repeatedly, with the actual value of probabilities referred to as *P*, predefined and fixed step sizes in the range of [0;1), expecting $step_{SH}<step_{H*}$; and the maximal value of hardest sampling probability $H{*}_{max}\in \left[0;1\right)$.
**Algorithm 1** Negative Sampling Probability Annealing**function** NSPA(P, $step_{SH}$, $step_{H*}$, $H{*}_{max}$)    $A\leftarrow [0,P\left[SH\right]+step_{SH},P\left[H*\right]+step_{H*}]$    $A\left[H*\right]\leftarrow \min A\left[H*\right],H{*}_{max}$    A[H]←1−A[SH]+A[H∗]    **for all** a∈A **do**        a←min{a,1}        a←max{a,0}    **end for**    **if** SUM(A)>1 **then**        A[ARGMAX(A)]←A[ARGMAX(A)]−(SUM(A)−1)    **end if**    **return** A**end function**

The algorithm increases the probability of semi-hard and hardest mining until P(H)=0, afterwards P(H∗) monotonically increases while P(SH) starts to decrease. A sample is shown on Figure 3.

Implementation and on-training use is detailed in Section 3.

## 3. Results

This section presents the results of the synthetic cluster analysis and the results of applying the proposed NSPA method. In the second part of the section, performance results of NSPA applied on a real-life dataset are given. The source codes and materials are publicly available; please refer to Appendix A.

### 3.1. Synthetic Cluster Analysis

To understand the effect of different negative sampling methods on cluster properties, a synthetic set of data points was generated. Data points were generated from 7 classes, each sample represented in a 32-dimensional space. During generation, for each class, a random point was selected in the space, and points near that center were generated based on Gaussian normal distribution. For training, 5 selected classes and a total of 1000 samples were selected, and 100 samples of each class were moved to a validation subset, and were not used directly for training, only to evaluate model performance.

Training was implemented in Keras [42] and Tensorflow [43] using a slightly modified version of the EmbeddingNet framework [44,45]. The embedding network used in the triplet architecture is a simple 32–16–2 fully connected network with ReLU activations. Adam optimizer [46] was used with a learning rate set to $5\cdot 10^{-4}$. No pretraining was applied. Mining was performed by selecting 20–20 samples from every 5 classes in each training step. A training iteration (or epoch) consisted of 50 training steps. Training concluded when loss measured on the validation set stopped improving for at least 25 epochs.

At every 10 epochs, embedded vectors of the validation dataset were collected and stored until training stopped.

A total of 20 training rounds were performed for each negative sampling method, resulting in a total of 60 logs of cluster information at different points of training. Figure A1 shows the results of the measurements, which were as expected: multiple studies reported that semi-hard negative sampling outperforms random hard sampling, and hardest sampling is suboptimal in every measured parameter.

Furthermore, the trainings showed that correlation was visible between the average distance between all elements, the average distance between the estimated centroids, and the average distance between the closest negatives. Similar behavior was observable in cases of calculating the average radius of clusters, the average distance of furthest positives, and the average distance between all positive pairs.

When evaluating the parameters normalized by the average element distance based on (Equation 20), it is visible (Figure A2) that the random negative sampling performs similarly to the semi-hard sampling, although with larger distances between the clusters.

When measuring performance, the percentage of negatives in a cluster and within marginal distance indicated the discriminative ability of the model. Results are visualized in Figure A3; it is clear that the hardest negative sampling struggled during the initial phases, and failed to improve.

The proposed NSPA method was implemented as a callback in Keras: at each epoch end, the probabilities were updated as defined in Algorithm 1. For the synthetic dataset, the values of $step_{SH}$, $step_{H*}$ and $H{*}_{max}$ were 0.1, 0.01, and 0.5, respectively.

Results of cluster analysis compared with the other methods are visualized in Figure 4. Further results are available in Appendix B.

Results on synthetic data are promising; NSPA performed similarly to semi-hard and random hard negative sampling, in some cases results are closer to hardest negative sampling.

To evaluate the performance, the proposed method was also applied to a benchmark dataset—NIST’19 [47].

### 3.2. Discriminative Ability

The NIST dataset is a wider alternative to the well-known MNIST dataset, which is actually a subset of it; the MNIST contains 10 classes of handwritten digits, the NIST also contains samples of handwritten letters.

While the MNIST samples are 28×28 images, 60,000 samples in total, NIST are 128×128 images in a total of 62 classes, which contains a sum of 814,255 samples.

Before applying the images, content highlighting preprocessing step was applied [23], and the resulting images were resized to 96×96.

The embedding neural network used was a simple convolutional neural network inspired by the VGG models [48]: after the 96×96 input, convolutional and pooling layers follow, and finally fully connected “dense” layers produced the output of 64 dimensions.

The marginal distance was set to 0.5; a maximum number of 1500 objects were selected for training per class. Adam optimizer [46] was used with a learning rate set to $10^{-4}$. Early stopping was applied after validation loss failed to improve for a number of 25 epochs. An epoch consisted of 20 steps, where each step 10 classes were selected and for each class 10 samples were used for mining. NSPA was applied with originally proposed values of 0.1, 0.01, and 0.5 for $step_{SH}$, $step_{H*}$, and $H{*}_{max}$, respectively. Probability updates happened after every 5 epochs.

The measured losses during training are visualized on Figure 5. This metric shows very similar behavior on all mining methods in this experiment.

For evaluation of discriminative ability, the *n*-way one-shot classification accuracy [49,50] was calculated for the proposed method and the other baseline sampling techniques. Measurements were performed on the training and on the validation subset as well. *n*-way one-shot classification accuracy calculation was based on distance comparison of an anchor object and a total of *n* objects, where n−1 elements were from different classes and exactly one was from the same class. If the distance for the positive pair was minimal, classification was correct; in other cases it was incorrect. The number of correct classifications in measurements gave the one-shot classification accuracy. Whichever method was used, it needed to outperform random classification, which was $\frac{1}{n}$ for *n* selected objects, i.e., 50% for two selected elements, and 25% for four samples. Table 1 presents the results of one-shot classification accuracy.

Each method was applied in training 5 times, resulting in a total of 20 trained models. Each models were evaluated using the one-shot classification accuracy, and the highest performances were selected in Table 1. Figure 6 visualizes the averages and standard deviations for classification performance of each method applied.

## 4. Discussion

In this paper, a cluster-analysis method was proposed to understand and measure embedded clustering dynamics. As an initial naïve approach, a synthetic set of multidimensional clusters were generated, and a triplet network with dimensionality-reduction goal was implemented. Different negative sampling methods were applied in multiple tests.

Results are as expected: semi-hard sampling outperforms other methods, with hardest sampling struggling at the initial phase of training. In the defined cluster properties, the average centroid distance and the average closest negative distance show high correlation: in both cases, the larger the distance, the better the performance.

In the case of the average distance between same-class elements—with average maximal positive distances and average cluster radii—a descending behavior was observed during training; however, in case of failed training rounds, the distances were short initially, and did not improve at any state. Thus, the values normalized with the average distance between elements provide further insight. Generally, we can conclude that a significantly reduced cluster radius has a negative effect on performance.

Clustering performance is measured by counting the number of different class elements inside a cluster, including marginal distance as well. Results show that most of the outliers are in case of hardest negative mining, where in some cases model was unable to fit, resulting to all negative embedded points appearing in all classes. It is also worth mentioning that the stability of the methods differs greatly, with the semi-hard sampling showing similar results on most training rounds.

When measuring the results for the proposed NSPA method, it is clear that its behavior is a mixture of all others, including average distances and performance as well. It is interesting that, in later phases—if reached—behavior is following the pattern of hardest sampling.

After analysis on synthetic data, the NIST dataset was used to benchmark performance. Discriminative ability is defined by one-shot classification accuracy. Results show that the best models perform similarly; however, NSPA and random hard mining seem to outperform the others.

It can be concluded that the proposed NSPA method is promising, although different settings should be evaluated, including lower maximum probability for hardest sampling and different limitations for random and semi-hard probability.

## 5. Conclusions

In this paper, a novel negative sampling solution is presented. Results indicate that the approach is promising; however, further research on hyperparameter tuning [51] could improve the applicability. It is worth mentioning that some experiments with suboptimal parameter settings showed similar behavior as the worst-performing setups.

Future goals include adaptation to multiple graphical accelerators to improve efficiency [52]; current online mining methods are computationally expensive, parallel, or distributed execution is limited.

While the so-called magnet loss [20] is computationally complicated, its advantages are undeniable. The aim of future research is to develop a method between random-sampling and step-by-step cluster analysis.

It is also worth mentioning that application of transfer learning is showing excellent results in deep metric learning, where applicable. Results reported in this paper should be analyzed in context of pretraining and knowledge transfer as well.

## Figures and Tables

**Figure 1 sensors-22-07579-f001:**
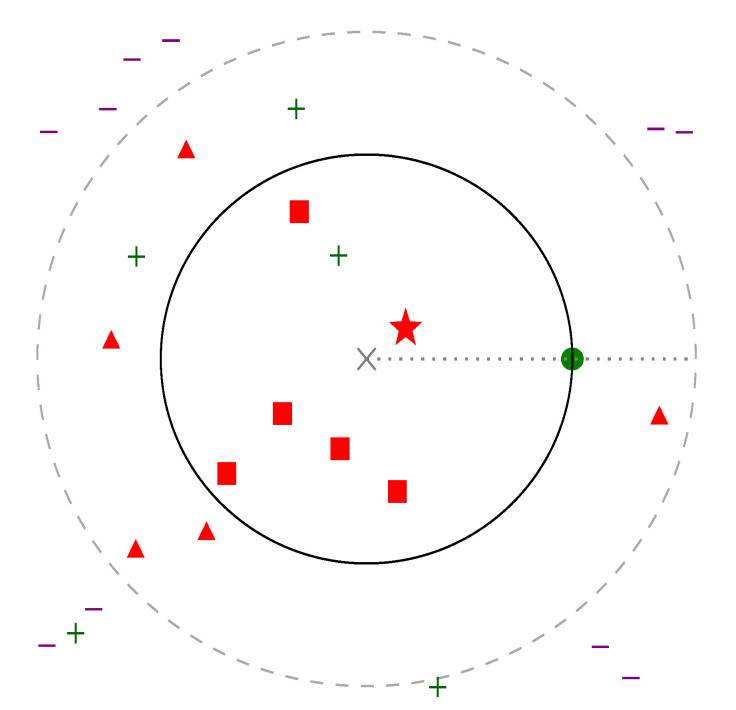
Two-dimensional representation of the embedded space. Gray cross in the middle marks the anchor element, *a*, and a green point shows a positive pair, *p*. $d^{+}$ positive distance is therefore defined and represented as a circle around *a*. Marginal distance ($d^{+}+m$) is similarly shown as a dashed circle. The closest negative sample, which provides the highest loss, is represented with a star; other elements where $d^{-}<d^{+}$ are shown with squares, and triangles represent negatives where $d^{+}<d^{-}<d^{+}+m$. Other positive and negative data points are represented with plus and minus signs, respectively.

**Figure 2 sensors-22-07579-f002:**
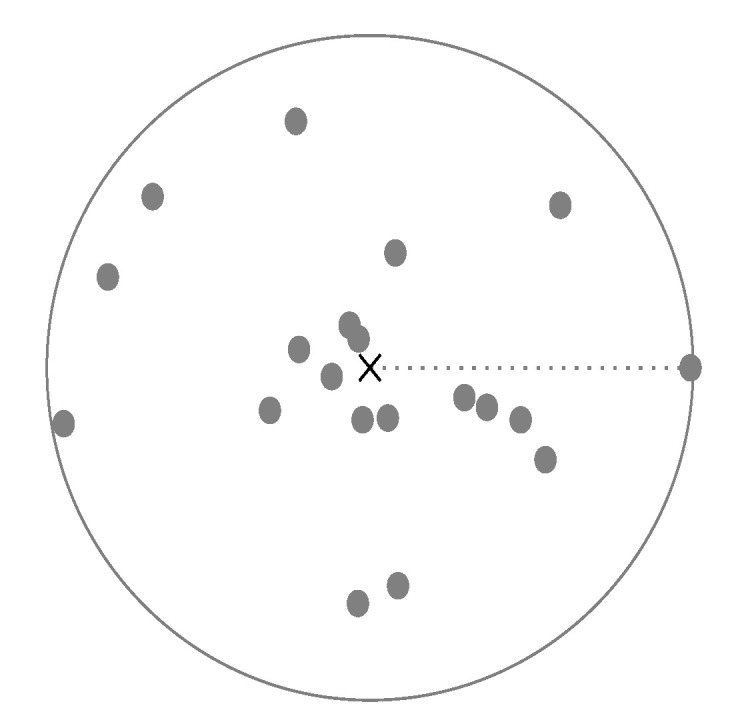
A visual representation of the centroid and the radius in two-dimensions. The centroid is the gravitational center of all data points of the same class, marked by a cross. The radius is given by selecting the maximal distance from the centroid to the data points, represented by a circle.

**Figure 3 sensors-22-07579-f003:**
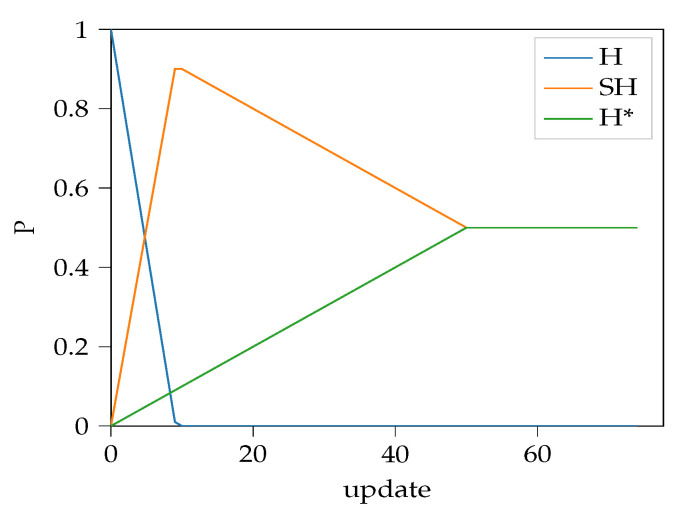
Sample behavior of the proposed probability annealing algorithm for $step_{SH}=0.1$, $step_{H*}=0.01$ and $H{*}_{max}=0.5$. Initial *P* is [1,0,0] for random hard, semi-hard, and hardest negative sampling, respectively.

**Figure 4 sensors-22-07579-f004:**
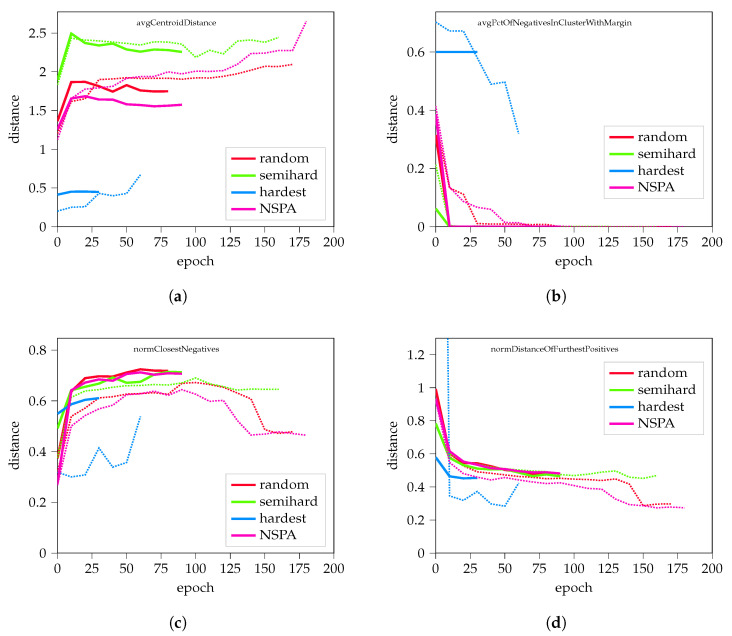
Results from applying the proposed NSPA method to synthetic data. (**a**) The average distance of cluster centroids; (**b**) the percent of different-class elements in the cluster; (**c**,**d**) display normalized results on the closest negative distance and the distance of the furthest positives, respectively. Dashed lines represent the mean values of different training experiments, while thick lines highlight representative records, which consist of average values including training length. Further results are available in the Appendix B, in Figure A4.

**Figure 5 sensors-22-07579-f005:**
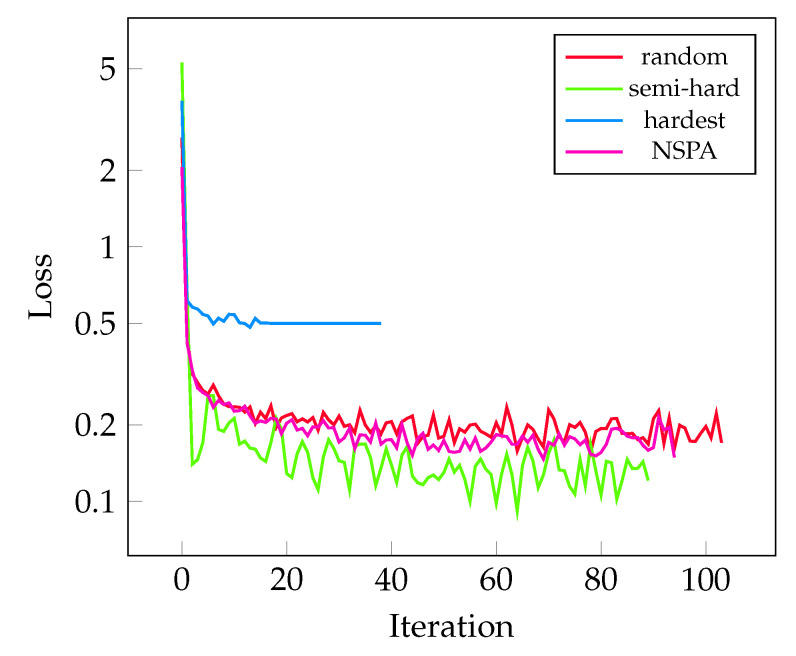
The improvement of the measured validation loss during training for different sampling methods. Each were tested in 5–5 training rounds; the best-performing models are selected here.

**Figure 6 sensors-22-07579-f006:**
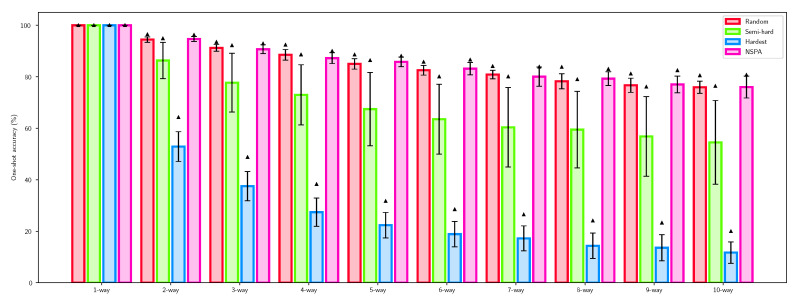
One-shot accuracy for *n*-way classification using different negative sampling techniques compared to the proposed NSPA algorithm. Error bars visualize the standard deviation of the performance of different models, while the triangle indicates the top accuracy.

**Table 1 sensors-22-07579-t001:** One-shot classification accuracy on the NIST dataset when applying different methods, values are given in percentage (%). Classification accuracy was measured for *n*-way, referring to the number of selected classes. Values in bold are the best for each column. Random classification is the chance of randomly selecting the correct class; it is only shown for reference.

Method	1-Way	2-Way	3-Way	4-Way	5-Way	6-Way	7-Way	8-Way	9-Way	10-Way
Random classification	100.0	50.0	33.3	25.0	20.0	16.7	14.3	12.5	11.1	10.0
Random hard negative mining	100.0	**96.5**	**93.5**	**92.4**	**88.6**	85.8	**84.1**	**83.8**	81.2	80.5
Semi-hard negative mining	100.0	94.9	92.2	88.6	86.4	80.1	80.1	79.0	76.1	76.4
Hardest negative mining	100.0	64.3	48.8	38.3	31.7	28.5	26.5	24.1	23.3	20.0
NSPA	100.0	94.8	92.9	90.0	88.1	**86.6**	83.2	83.0	**82.5**	**80.7**

## Data Availability

Not applicable.

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
