# Peer review of "Deep Metric Learning Using Negative Sampling Probability Annealing"

_sensors, 2022, doi:10.3390/s22197579_

Round 1

Reviewer 1 Report

In this paper, the authors present the NSP technique to dynamically select semi-hard negative mining or hardest mining to complete the task of sample selection in deep metric learning. Lots of theory analysis and experimental validation are provided to confirm the advantage of this method. Overall, the study is a good work, and it is both complete and convincing. I have the following concerns:

1. Abstract: It’s better to add the experimental analysis in this part. It’s better to point out the main problem to be dealt with.

2. Introduction: The type of metric learning based on Riemannian manifold [1-3] can be added to enrich this part, and more basic knowledge about manifold can be referred to [3].

[1] R. Wang, X. -J. Wu and J. Kittler, "Graph Embedding Multi-Kernel Metric Learning for Image Set Classification With Grassmannian Manifold-Valued Features," in IEEE Transactions on Multimedia, vol. 23, pp. 228-242, 2021, doi: 10.1109/TMM.2020.2981189.

[2] M. Dai and H. Hang, "Manifold Matching via Deep Metric Learning for Generative Modeling," 2021 IEEE/CVF International Conference on Computer Vision (ICCV), 2021, pp. 6567-6577, doi: 10.1109/ICCV48922.2021.00652.

[3] X. Hua, Y. Ono, L. Peng and Y. Xu, "Unsupervised Learning Discriminative MIG Detectors in Nonhomogeneous Clutter," in IEEE Transactions on Communications, vol. 70, no. 6, pp. 4107-4120, June 2022, doi: 10.1109/TCOMM.2022.3170988.

3. It’s better to add some references for the existing equations.

4. It’s better to summary the main contribution in the introduction part.

5. Conclusion: The authors missed the potential drawbacks of the proposed method.

Reviewer 2 Report

Review of the manuscript:

 Deep Metric Learning using Negative Sampling Probability Annealing

In this manuscript introduces a technique of dynamic policy switching, referred to as Negative Sampling Probability Annealing, which aims to exploit the positives of all approaches. A method of cluster analysis is introduced, based on the observations a novel negative sampling method is proposed and evaluated.

1. The findings are sufficiently novel to warrant publication.

2. The conclusions are adequately supported by the data presented.

3. The article is clearly and logically written so that it can be understood by one who is not an expert in the specific field.

4. The work provides an important contribution to its field, consistent with the scope of the journal, the problematics is clearly described. Methods are described and but not explained.

Comments:

Rows 109 – 111: Please introduce the literature to the equations (1) – (11)

Row 112: Please introduce the literature about cluster analysys

Rows 116 -132: Please introduce the literature to the equations (12) – (20)

Row 174: Please introduce the applied software and shortly describe the minig method. All essential steps you have described only by the literature citation.

Round 2

Reviewer 1 Report

The authors have addressed all my concerns.